# Metagenomic Analysis Reveals A Gut Microbiota Structure and Function Alteration between Healthy and Diarrheic Juvenile Yaks

**DOI:** 10.3390/ani14081181

**Published:** 2024-04-14

**Authors:** Hongwen Zhao, Quan Mo, Muhammad Fakhar-e-Alam Kulyar, Jiuqiang Guan, Xiangfei Zhang, Xiaolin Luo, Jiakui Li

**Affiliations:** 1Sichuan Academy of Grassland Sciences, Chengdu 611731, China; zhwdjc2014@163.com (H.Z.); gjq1900@163.com (J.G.); zxfsicau@foxmail.com (X.Z.); 2College of Veterinary Medicine, Huazhong Agricultural University, Wuhan 430070, China; moquan@mail.hzau.edu.cn (Q.M.); fakharealam786@hotmail.com (M.F.-e.-A.K.)

**Keywords:** metagenomic analysis, yaks, microbiota, 16S rRNA, diarrhea

## Abstract

**Simple Summary:**

Diarrhea is a major cause of mortality in young yaks. This study investigates its impact on the gut microbiota of these animals. The results show a decrease in microbial diversity and an increase in harmful bacteria like Eysipelatoclostridium, Parabacteroides, and Escherichia-Shigella during diarrhea. A metabolic pathway analysis reveals heightened activity in the associated pathways. These outcomes indicate that dysbiosis in the gut microbiota may be a key factor driving diarrhea outbreaks in yaks in the Qinghai-Tibet plateau.

**Abstract:**

Diarrhea-induced mortality among juvenile yaks is highly prevalent in the pastoral areas of the Qinghai-Tibet plateau. Although numerous diseases have been linked to the gut microbial community, little is known about how diarrhea affects the gut microbiota in yaks. In this work, we investigated and compared changes in the gut microbiota of juvenile yaks with diarrhea. The results demonstrated a considerable drop in the alpha diversity of the gut microbiota in diarrheic yaks, accompanied by *Eysipelatoclostridium*, *Parabacteroides*, and *Escherichia-Shigella*, which significantly increased during diarrhea. Furthermore, a PICRust analysis verified the elevation of the gut–microbial metabolic pathways in diarrhea groups, including glycine, serine, and threonine metabolism, alanine, aspartate, oxidative phosphorylation, glutamate metabolism, antibiotic biosynthesis, and secondary metabolite biosynthesis. Taken together, our study showed that the harmful bacteria increased, and beneficial bacteria decreased significantly in the gut microbiota of yaks with diarrhea. Moreover, these results also indicated that the dysbiosis of the gut microbiota may be a significant driving factor of diarrhea in yaks.

## 1. Introduction

The gut microbiota is vital for animal health in differentiating and proliferating intestinal epithelial cells and regulating immune responses [1,2]. It helps to preserve the integrity of the intestinal barrier by promoting the maturation and maintenance of intestinal epithelial cells [3]. The gut microbiota also serves as a regulator, promoting a harmonized and tolerant reaction to harmless antigens, and it impacts the maturation and functionality of immune cells, molding the host’s immunity [4]. Moreover, the gut microbiota influences the host’s growth and development, especially in the early stages of life. It helps nutrient absorption, hormone balance, and the maturation of critical physiological systems like the immunological and neurological systems [5]. The metabolic functions of the gut microbiota involve the breakdown and fermentation of dietary components that the host cannot independently digest, such as complex carbohydrates. Through these metabolic activities, the gut microbiota generates essential nutrients, short-chain fatty acids, and other bioactive molecules, thereby influencing the metabolism and energy balance of the host [6]. Simultaneously, certain gut microbes can produce bacteriostatic compounds, hindering the growth of pathogenic bacteria. The intestinal microbiota’s resilience is critical for the intestine to operate as a mechanical and immunological barrier against harmful microbes [7]. Nonetheless, the temperature, environment, medications, and host phenotypes can all impact the intestinal microbiota’s stability [8,9]. Previous studies have shown a strong link between gut microbiota changes and various illnesses, including enteritis, obesity, diabetes, and diarrhea [10,11]. 

Diarrhea is a primary cause of decreased production performance and mortality in livestock, substantially influencing the livestock industry’s development worldwide [12]. Diarrhea is mainly attributed to infectious and non-infectious factors, but regardless of which factor causes diarrhea, it can disrupt the host’s gut microbiota. Extensive research has highlighted the prevalence of diarrhea across various mammalian species, particularly affecting vulnerable gut microbiota in lambs and piglets [13,14,15]. An imbalance in the gut microbiota has been linked to numerous gastrointestinal disorders, including diarrhea and irritable bowel syndrome. Many studies have underscored the gut microbiota’s pivotal role in preventing, controlling, and diagnosing diarrhea [16,17]. The gut microbiota plays a protective role in pathogen defense, as dysbiosis of the intestinal microbiota can affect various microbial functions related to nucleotide transport and metabolism, defense, translation, and transcription [18]. Furthermore, reduced gastrointestinal barrier function and disruptions in the gut microbiota are associated with diarrhea and alterations in the gut microbiota [19]. Moreover, diarrhea and gut microbiota disruptions have a bidirectional relationship [20]. Specifically, diarrhea disrupts the intestinal microbiome balance, while an imbalance in the intestinal microbiota due to high-value-added exogenous pathogens can also lead to diarrhea. Diarrheic calves exhibited reduced gut microbiota diversity and significant changes in fecal microbial composition compared to healthy calves [21]. Wang et al. revealed notable changes in the intestinal microbiota of diarrheal goats, together with higher death rates [15]. Similarly, Li et al. discovered significant alterations in giraffes’ intestinal microbiome after episodes of diarrhea [22]. Furthermore, diarrhea is characterized by a decrease in beneficial bacteria that produce short-chain fatty acids (SCFAs), which play a role in reducing the risk of diarrhea.

Yaks primarily inhabit high-altitude, low-oxygen environments and have evolved to possess unique digestive features and gut microbiota [23,24]. However, immature gastrointestinal development and unstable gut microbiota in juvenile yaks, coupled with the harsh natural conditions and extensive traditional breeding practices in the Qinghai-Tibet plateau, lead to a high incidence of diarrhea, resulting in a high fatality rate [25]. Thus, in contrast to poultry and other mammals, the importance of the gut microbiota in numerous physiological activities is more prominent in yaks. Zhang et al. demonstrated that the predominant bacterial phyla in the guts of healthy yaks were Firmicutes, Bacteroidetes, and Proteobacteria. A substantial presence of anaerobes from families such as Peptostreptococcaceae, Prevotellaceae, Flavobacteriaceae, Lachnospiraceae, and Succinivibrionaceae was observed across different intestinal segments [23]. The analysis of fungal microbial diversity revealed significant differences in 23 genera, including Paraconiothyrium, Monascus, Plenodomus, Phaeoacremonium, Preussia, and Mortierella, between healthy and diarrheal adult yaks from the Gannan region of the Gansu Province, China [26]. Furthermore, the examination of the intestinal bacteria indicated a lower relative abundance of Firmicutes, Bacteroidetes, and Verrucomicrobia in diarrheal adult yaks compared to their healthy counterparts [27]. Nevertheless, the specific composition and functional alterations of the intestinal bacteria in diarrheic yaks, particularly juvenile calves, remain unclear. The neonatal period in yak calves is critical for establishing the gut microbial community, as the composition formed during this period significantly influences the structure of the microbiota in later adulthood. Importantly, the gut microbiota’s configuration during this stage is highly adaptable, with enduring effects on gut function, including susceptibility to diarrhea, extending into later life. Hence, this study investigates the alterations in gut microbiota structure and function between healthy and diarrheic juvenile yaks, offering insights that could inform the development of preventive and therapeutic strategies for diarrhea from a gut microbial perspective.

## 2. Materials and Methods

### 2.1. Animals and Sample Collection

A total of eight healthy and eight diarrheal yaks from the Hongyuan Yak Breeding Base in Sichuan, China (located in the eastern part of the Qinghai-Tibet plateau at an altitude above 3500 m) were selected, having the same immunological history. All yaks were vaccinated against Brucella and foot-and-mouth disease but not for enterotoxigenic Escherichia coli and bovine coronavirus antibodies. All yaks were provided with the same amount of green grass and a daily supplement of 0.5 kg of purified diet (corn meal, bran, soybean meal, NaCHO_3_, etc., in proportion to calves’ nutritional requirements). Before collecting the samples, professional veterinarians evaluated and analyzed the health statuses of the yaks and determined that pathogenic infections did not cause diarrhea in the yaks and had not been treated. When the diarrheal group presented clinical signs of diarrhea, the rectum was swabbed with sterile swabs in a rotating way. The obtained samples were immediately placed in sterile plastic containers, transported to the laboratory, and stored at −80 °C for future investigation.

### 2.2. Microbiome Sample Processing and Sequencing

According to the manufacturer’s instructions, genomic DNA was isolated from the rectal contents using the QIAamp Fast DNA Stool MiniKit (Qiagen, Inc., Hilden, Germany). The genomic DNA’s concentration and purity were assessed using a 1% agarose gel. The V3-V4 region of the bacterial 16S rRNA gene was amplified using universal primers (338F, 5′-ACTCCTACGGGAGGCAGCA-3′; and 806R, 5′-GGACTACHVGGGTWTCTAAT-3′) [28]. PCR amplification was performed over 25 cycles at an annealing temperature of 57 °C. The PCR products were confirmed using 1% gel electrophoresis, then cleaned and normalized with the SequalPrep Normalization Plate Kit (Life Technologies, Carlsbad, CA, USA). Following the manufacturer’s instructions, the purified amplification PCR products were transformed into a sequencing library using the Next Ultra DNA Library Prep Kit (New England BioLabs [NEB], Ipswich, MA, USA). After quality testing with a bioanalyzer (Agilent Technologies, Santa Clara, CA, USA) and quantitative PCR (qPCR), only the libraries with a single peak and a concentration greater than 2 nM were selected for high-throughput sequencing. The qualifying library was subsequently sequenced using the Illumina HiSeq 6000 platform (San Diego, CA, USA), emphasizing paired-end reads.

### 2.3. Bioinformatics and Statistical Analysis

The demultiplexing and clipping of the sequence adapters from the raw sequences were performed using CASAVA data analysis software (Illumina; v1.8.2). Paired-end sequences were merged using PEAR v0.9.1016 with the default parameters. Subsequently, the sequences with an average quality score lower than 20 and containing unresolved bases were removed with the split_libraries_fastq.py script from QIIME 1.9.117. Further, we removed the non-clipped reverse and forward primer sequences by employing cutadapt 1.1018 with the default settings. The reads that successfully passed the quality evaluation underwent clustering and operational taxonomic unit (OTU) discrimination at a 97% similarity threshold. The representative sequence of each OTU was taxonomically classified based on the ribosomal database project (RDP) database. The phylogenetic relationships of different OTUs and multiple sequence alignments were conducted using MUSCLE software v5. 

Furthermore, Venn diagrams were generated to determine the number of shared operational taxonomic units (OTUs) between the two groups. The alpha diversity analysis, which included the Shannon, Simpson, Chao1, and ACE indexes, was calculated using Mothur. The beta diversity based on the weighted UniFrac distance matrices was calculated with QIIME (Version 1.7.0), and a cluster analysis was preceded by a principal coordinate analysis (PCoA). A metastats analysis and LEfSe were employed to identify differentially abundant bacteria at various taxonomic levels. A heat map was created via R software (v3.0.3). The data were evaluated statistically via a one-way analysis of variance. 

Based on the species abundance, each genus’s correlation coefficient (Spearman correlation coefficient) was calculated, and the correlation coefficient matrix was obtained. Cytoscape V. 3.6.1 (https://cytoscape.org/, accessed on 25 January 2024) was used to visualize the networks with significant correlations between genera.

### 2.4. Functional Profile Analysis of the Intestinal Bacterial Community via PICRUSt

PICRUSt software V. 1.1.4 (accessed on 26 January 2024) was used to predict the functionality of the various metagenomes categorized by sample type. The KEGG ortholog function of the bacteria micropopulation in yaks was predicted using the PICRUSt annotating databases of KEGG and COG [29]. Alterations in the function of the intestinal bacteria were analyzed through ANOVA and Dunn tests.

### 2.5. Statistical Analysis

The amplicon sequencing data were analyzed using GraphPad Prism (v8.0) and the SPSS statistical tool (v20.0). The probability (*p*) values (means ± SD) < 0.05 indicated statistical significance. The values were corrected using a false discovery rate (FDR) analysis for the KEGG pathways. All statistical tests were two-tailed, denoting *p* values ≤ 0.05 as * and *p* values ≤ 0.01 as **. 

## 3. Results

### 3.1. Sequence Analyses

The current microbiome study used amplicon sequencing to analyze 16 fecal samples from healthy and diarrheal yaks. The V3/4 areas yielded 122,811.2 original sequences (control group = 605,722, diarrhea group = 622,390). Following sequence filtering, a total of 1,205,994 samples (control group = 594,104, diarrhea group = 61,950) were obtained, with an average of 75,374 reads per sample (with a range of 56,588 and 79,196 reads) (Table 1). The rarefaction curves (Chao1 curve, Shannon curve) and rank abundance curves tended to saturate, indicating suitable depth and evenness (Appendix A). Ninety-seven percent of nucleotide sequence similarity among the high-quality sequences was used to identify an OTU. There were 3608 OTUs found in the gut bacterial communities, with numbers ranging from 292 to 1549 in each sample (Appendix A). In addition, the healthy and diarrheal yaks had 2102 and 2105 OTUs, respectively, and 599 OTUs in common, making up roughly 16.60% of the total OTUs (Appendix A), suggesting that there were considerable variations in the intestinal bacterial populations between the healthy and diarrheal yaks.

### 3.2. Analysis of the Microbial Diversity in the Healthy and Diarrheic Yaks

To better analyze the variations in the gut bacterial populations in diarrheal yaks. Chao1, ACE, Shannon, and Simpson tests were employed to assess the alpha diversity of the gut microbiome community. The average Shannon index was 6.87 and 6.29 in the healthy and diarrheal yaks, respectively. The average Chao1 and ACE indexes in the control group were 456.31 and 410.80, whereas those in the diarrheic group were 413.00 and 391.20, respectively (Figure 1A–C). Additionally, the diarrheic group had an average Simpson index of 6.41, and the control group had an average of 5.861 (Figure 1D). The statistical analysis indicated that the diversity indexes of the healthy yaks, including Chao1, ACE, Shannon, and Simpson, were considerably larger than those of diarrheic yaks. This suggests a significant difference in the richness and diversity of the gut flora community between healthy and diarrheic yaks. Moreover, the similarity and variability between the intra-group and inter-group samples were assessed using a PCoA. The gut microbiota PCoA scatterplot separated the samples between the diarrheic and healthy yaks (Figure 1E), suggesting that diarrhea significantly impacted the yaks’ primary gut microbial assemblages.

### 3.3. Composition Analysis of the Gut Microbial Community in the Healthy and Diarrheic Yaks

Both healthy and diarrheagenic yaks were examined at different taxonomic levels to determine the composition of the gut microbial population. According to the findings, 20 phyla were identified in the gut bacterial population, with each sample containing 17–20 phyla. In the healthy yaks, the most prevalent phyla were Firmicutes (81.58%), Bacteroidota (10.71%), Desulfobacterota (3.97%), and Campylobacterota (1.13%), which together consisted of a total 97.39% of the bacterial composition. However, among the diarrhea groups, Bacteroidota (51.26%) was the most prevalent bacterial phylum, followed by Firmicutes (35.74%), Proteobacteria (6.62%), and Desulfobacterota (2.25%), which accounted for roughly 95.87% of all bacterial taxa (Figure 2A). In addition, Methylomirabilota, Chloroflexi, and unclassified_Archaea were only detected in the control group. In contrast, Phyllovibrionota was detected only in the diarrhea group. To further evaluate the changes at the genus level of the bacterial composition in the gut during diarrhea, a total of 34 genera were identified (Figure 2B). Among these, the relative abundance of *unclassified_Lachnospiraceae* (7.0% vs. 32.76%), *Lactobacillus* (2.34% vs. 13.18%), *Lachnospiraceae_NK4A136_group* (3.71% vs. 9.02%), unclassified_Desulfovibrionaceae (1.80% vs. 3.58%), *unclassified_Bacilli* (1.04% vs. 4.23%), *Blautia* (0.54% vs. 1.86%), *Enterorhabdus* (0.64% vs. 0.91%), *Roseburia* (0.36% vs. 1.16%), *Colidextribacter* (0.27% versus1.14%), and *Bacillus* (0.02% vs. 1.15%) in the diarrheal group were less than those in the control group. The relative abundance of *unclassified_Muribaculaceae* (23.00% vs. 4.19%), *Alloprevotella* (10.51% vs. 0.83%), *Ligilactobacillus* (5.89% vs. 5.12%), *Bacteroides* (5.32 vs. 0.37%), *Alistipes* (2.72% vs. 2.34%), *Parasutterella* (4.85% vs. 0.25%), *uncultured_Bacteroidales_bacterium* (3.76% vs. 0.97), *Turicibacter* (1.26% vs. 0.28%), *Erysipelatoclostridium* (1.09% vs. 0.36%), *Dubosiella* (1.35% vs. 0.07%), *Parabacteroides* (1.24% vs. 0.11%), and *Prevotellaceae_UCG_001* (1.08% vs. 0.19%) in the diarrheal group were more than in the control group. Additionally, the heatmap shows the heterogeneity of the gut bacterial population in diarrheagenic yaks and the distribution of the identified bacterial genus (Figure 2C).

A metastats analysis was performed at several taxonomic levels to better understand how the gut microbiota of yaks changes during diarrhea. The phylum-level analysis revealed that the diarrheic yaks had significantly higher bacteroidota and proteobacteria abundances than the control group, while the diarrheic yaks had significantly lower Firmicutes abundance. Furthermore, compared with the control group, the relative abundances of 22 bacterial genera (*unclassified_Muribaculaceae*, *uncultured_Bacteroidales_bacterium*, *Parasutterella*, *Parabacteroides*, *Rikenellaceae_RC9_gut_group*, *uncultured_Muribaculaceae_bacterium*, *Muribaculum*, *Dubosiella*, *Escherichia_Shigella*, *Streptococcus*, *UCG_005*, *unclassified_Anaerovoracaceae*, *unclassified_Clostridia*, *unclassified_UCG_010*, *Rikenella*, *Family_XIII_UCG_001*, *Bifidobacterium*, *Faecalibaculum*, *Allobaculum*, *Frisingicoccus*, *Cetobacterium*, and *Coriobacteriaceae_UCG_002*) were dramatically increased, whereas the relative abundances of 29 bacterial genera including *unclassified_Lachnospiraceae*, *Lactobacillus*, *Lachnospiraceae_NK4A136_group*, *unclassified_Bacilli*, *Alloprevotella*, *Colidextribacter*, *Bacillus*, *[Eubacterium]_xylanophilum_group*, *Lachnospiraceae_UCG_006*, *[Eubacterium]_siraeum_group*, *Bacteroides*, *unclassified_Peptococcaceae*, *Rikenella*, *Family_XIII_UCG_001*, *[Eubacterium]_ruminantium_group*, *unclassified_Cyanobacteriales*, *Peptococcus*, *unclassified_UCG_010*, *Pediococcus*, *[Acetivibrio]_ethanolgignens_group*, *unclassified_Anaerovoracaceae*, *UCG_003, [Eubacterium]_oxidoreducens_group*, *Limosilactobacillus*, *[Eubacterium]_ventriosum_group*, *Leuconostoc*, *Fusobacterium*, *unclassified_Lactobacillales*, *unclassified_Beijerinckiaceae*, *Candidatus_Udaeobacter*, and *Dialister* were significantly decreased (Appendix A). Since this discriminant analysis may not have detected the entire taxonomic composition, LEfSe, in conjunction with the LDA scores, was used to identify the taxonomic compositions at the genus level among the groups. We also found that the relative abundances of *Lactobacillus*, *Eubacterium oxidoreductase group*, *unclassified_Bacilli*, *unclassified_Lachnospiraceae*, and *Lachnospiraceae_NK4A136_group* were observed to be predominant in the control group, whereas the relative abundances of *Peptococcus*, *uncultured Bacteroidales bacterium*, *Bacteroides*, *Alloprevotella*, *unclassified Muribaculaceae*, and *Parasutterella* were the most dominant in the diarrheal group (Figure 3A,B).

### 3.4. Correlation Network Analysis

Lachnospiraceae_NK4A136_group was positively associated with *Colidextribacter* (0.81), unclassified_Peptococcaceae (0.84), Lachnospiraceae_UCG_006 (0.74), *Oscillibacter* (0.74), and unclassified_Desulfovibrionaceae (0.73). Additionally, unclassified_Muribaculaceae was positively associated with uncultured_Bacteroidales_bacterium (0.92), *Muribaculum* (0.84), Bacteroides (0.78), Parabacteroides (0.78), and unclassified_Lachnospiraceae, which was negatively associated with *Allobaculum* (−0.75), Bacteroides (−0.81), *Faecalibaculum* (−0.78), Parabacteroides (−0.74), *Parasutterella* (−0.81), and unclassified_Muribaculaceae (−0.80). *Turicibacter* was positively associated with *Allobaculum* (0.74), *Bifidobacterium* (0.76), Christensenellaceae_R_7_group (0.79), Dubosiella(0.78), *Faecalibaculum* (0.70), *Romboutsia* (0.80), and *Parasutterella* (0.83). Coriobacteriaceae_UCG_002 was positively associated with *Allobaculum* (0.76), *Bifidobacterium* (0.80), *Streptococcus* (0.75), and unclassified_Atopobiaceae (0.75) (Figure 4).

### 3.5. Functional Predictions of the Intestinal Microbes in the Diarrheic Group and Healthy Group

A PICRUSt metagenomic functional prediction was performed to connect the microbial genes to the KEGG metabolic database. This was done to compare the functional capability of the mucosal taxa between the two groups. Thirteen KEGG pathways with substantial changes between the two groups were identified (Figure 5). The diarrhea group showed highly enriched pathways, i.e., metabolism, carbon fixation in prokaryotes, the biosynthesis of antibiotics, oxidative phosphorylation, the biosynthesis of secondary metabolites, glycine, serine, and threonine metabolism, alanine, aspartate, and glutamate metabolism, whereas the control group showed considerably enriched pathways of ABC transporters, quorum sensing, and the pentose phosphate pathway.

## 4. Discussion

Plateau yak farmers still suffer significant productivity and financial losses due to calf diarrhea, a disease frequently recorded in juvenile animals. Statistics have indicated that diarrhea is the cause of half of the mortality amongst unweaned yaks [26,30]. However, several elements, including a harsh climate, malnutrition, and stress reactions, make controlling diarrhea in yaks very challenging [31]. Extensive studies have shown that the disruption of the intestinal microbiota is a significant factor leading to diarrhea [13,15]. The gut microbiota is a vital defense system, offering protection against foreign pathogen invasion and colonization while also playing an important role in illness prevention and therapy [16]. Consequently, the investigation of the gut microbiota has attracted widespread attention. Nevertheless, limited studies have explored the gut microbiota in juvenile yaks with varying health statuses. This work evaluated the gut microbial composition of healthy and diarrheal yaks. The findings unveiled a notable contrast in the bacterial abundance and diversity between the two groups.

Compared to the healthy group, we observed a significant reduction in the variety and richness of the gut microbiota in the diarrheal yaks. This indicates the presence of gut bacterial dysbiosis. Consistent with our investigations, Li et al. profiled the fecal microbial community of 21 calves with varying health conditions using the 16S rRNA gene and also found that the diversity and evenness index of calves with diarrhea were significantly reduced [32]. The detection of the gut microbiota structure of adult yaks with diarrhea also presents a distinct [27] line in alpha diversity, accompanied by significant shifts in taxonomic compositions. In addition, Wang et al. found that diarrheic Boer goats had much lower gut bacterial diversity than healthy herds [15]. Furthermore, giraffes’ gut bacterial diversity decreased during diarrhea [22]. Juvenile animals’ gut microbiome community structures are volatile and quickly change by species, genotype, environment, and nutrition during development, eventually stabilizing at maturity [33,34]. Research has shown that maintaining a proper gut bacteria composition and diversity is necessary for performing complicated physiological processes. Conversely, intestinal microbiome disturbance is identified as a central or driving factor in various diseases, including enteritis and diarrhea [34]. Dysbiosis of the intestinal microbiota has been connected to diarrheal yaks, marked by increased mortality and weight loss. 

Previous research suggests that gut microbial dysbiosis can affect human immunity and intestinal permeability, increasing vulnerability to pathogens. Additionally, specific pathogens may exhibit heightened pathogenicity in the context of intestinal flora disturbance. Consequently, we speculate that the increased mortality and weight loss in diarrheic yaks may be related to gut microbial dysbiosis. This study found that Firmicutes and Bacteroidetes were the most common bacterial phyla in the intestines of a healthy yak calf. In line with our results, the prevalence of these two phyla was noted in the intestinal tracts of sheep, cattle, and steers, underscoring the ecological and functional significance of these dominant phyla in the gastrointestinal tract of ruminants [35,36,37]. Firmicutes encompass many Gram-positive bacteria, including beneficial species such as *Lactobacillus*, *Lactococcus*, and *Listeria*. These bacteria are crucial in maintaining a balanced gut microbiota and preventing pathogenic invasions [38,39]. In ruminants, Firmicutes is mainly involved in the digestion of fiber and cellulose, contributing to overall digestive processes [40].

On the other hand, Bacteroidetes primarily handles the digestion of carbohydrates and proteins, playing a beneficial role in the maturation of the intestinal immune system [41]. However, diarrheic calf yaks show a significant decrease in the proportion of Firmicutes and Bacteroidetes, while the abundance of Proteobacteria is notably elevated. A similar result was also found in adult yaks with diarrhea: the relative abundance of Proteobacteria increased, while the relative abundance of Bacteroidetes significantly decreased [27]. Similarly, Proteobacteria (mostly Enterobacteriaceae/*E. coli*) was most enriched during this early phase of human diarrhea and could account for up to 60% of the relative abundance of the fecal microbiomes [27]. Proteobacteria harbors many Gram-negative pathogenic bacteria, including *Vibrio cholerae*, *Salmonella* spp., *Helicobacter pylori*, and *Escherichia coli*. An elevated abundance of *Proteobacteria* in the intestines raises the susceptibility to pathogenic infections [42]. Moreover, an expansion in the Proteobacteria population leads to increased lipopolysaccharide (LPS) production, which can lead to intestinal inflammation in the host [43]. Therefore, the low Firmicutes, Bacteroidetes, and high *Proteobacteria* content of diarrheic yaks could be among the causes of limited growth, increased mortality, and weight loss.

In the present study, substantial alterations were identified within specific bacterial genera during instances of diarrhea. These observations underscore the potential pivotal roles that these genera might play in influencing the equilibrium of the gut microbiota and contributing to the onset and development of diarrhea. Furthermore, some bacterial genera in the diarrheal yaks showed a sharp decline, including *Lactobacillus*, *Lachnospiraceae_NK4A136_group*, *Blautia*, *Enterorhabdus*, *Roseburia*, *Colidextribacter*, and *Bacillus*, which are regarded as beneficial intestinal bacteria and are essential for intestinal functioning and host health. *Lactobacillus* has been widely recognized as an advantageous gut bacterium, possessing favorable biological traits like promoting growth, enhancing immunity, and preserving microecological balance [44]. Furthermore, *Lactobacillus* demonstrates antibacterial effects by generating organic acids antimicrobial peptides and competing with pathogenic bacteria for adhesion sites [45]. Zhang et al. discovered that *Lactobacillus* decreased the number of pathogens in the intestines of yaks infected with *Escherichia coli* O78 and raised the number of beneficial gastrointestinal microbiota in yaks [46]. Li et al. tracked the fecal microbiota composition of a calf during the incubation stage, prodromal stage, and recovery phases of diarrhea. They showed that the abundance of Lactobacillus increases and returns to baseline levels early in the recovery phase. The structure and composition of the fecal microbiota in diarrheic calves exhibited a higher abundance of Lactobacillus, suggesting that Lactobacillus may be a marker of intestinal microbiota changes during the recovery period of diarrhea [32].

The Lachnospiraceae NK4A136 group, identified as a potential probiotic in the rumen and intestine, negatively correlates with intestinal inflammation [47]. This group has been previously recognized for its capacity to produce short-chain fatty acids (SCFAs), which positively regulate the physiological functioning of the intestines and gut permeability. In the early phase of human diarrhea, the drastic disappearance of Blautia and Lachnospiraceae NK4A136 leads to a depletion of associated metabolites such as short-chain fatty acids (SCFAs) [32,48]. Recent studies have suggested that the Lachnospiraceae NK4A136 group demonstrates potential anticolonic inflammatory activity [49]. *Blautia* bacteria are acknowledged as intestinal microbes capable of producing butyrate. *Blautia* bacteria play a role in regulating glucose metabolism disorders and inflammation associated with obesity [50]. Furthermore, increasing *Blautia* abundance helps to reduce intestinal permeability, non-alcoholic fatty liver, and liver cirrhosis [51]. *Enterorhabdus* is a primary bacterial genus producing bile salt dehydrogenase and deconjugating bile acids (BAs). Through the 7α-dehydroxylation process, *Enterorhabdus* transforms unconjugated primary bile acids (such as CA and CDCA) into secondary bile acids (such as DCA and LCA) [52]. Additionally, *Enterorhabdus* negatively correlates with the concentration of pro-inflammatory cytokines, positively maintaining intestinal homeostasis [53]. *Colidextribacter* and *Roseburia* can also produce short-chain fatty acids (SCFAs), which can control intestinal permeability and preserve the gut’s regular physiological processes [54]. *Bacillus* is involved in tryptophan metabolism, maintaining intestinal barrier integrity, and influencing host immune responses [55]. The stability of beneficial bacteria in diarrheic yaks is crucial for host health and intestinal balance. Thus, the decline in these bacteria may trigger diarrhea in yaks. Under normal physiological conditions, gut microbes form commensal, synergetic, or antagonistic relationships to maintain intestinal balance. The correlation network analysis of this study shows a noteworthy correlation between reduced beneficial bacteria and other species, suggesting diarrhea might affect additional bacteria through interactions, impact the gut microbial community, and exacerbate dysbiosis.

Additionally, it is noteworthy that in the diarrheic group, the abundances of potential pathogenic bacteria, i.e., *Bacteroides*, *Turicibacter*, *Parasutterella*, *Eysipelatoclostridium*, *Parabacteroides*, and *Escherichia-Shigella* were significantly increased. *Bacteroides*, a conditional pathogen, has been reported to be highly enriched in mice with colitis [56]. Within *Bacteroides*, enterotoxigenic Bacteroides fragilis (ETBF) increases intestinal epithelial cell permeability and disrupts epithelial barrier function, leading to acute or chronic intestinal inflammation [57]. *Turicibacter* is an inflammation-promoting bacterium that can induce enteritis and sepsis as host immunity decreases [58]. *Parasutterella*, a strictly anaerobic Gram-negative coccus from the phylum Bacteroidetes, has been reported to cause gut dysbiosis, a diminished variety of gut microbiota, leading to the eventual emergence of intestinal or metabolic disorders such as inflammatory bowel disease and obesity [59]. *Eysipelatoclostridium* is intricately linked to intestinal toxemia and diarrhea in mammals, with its toxins impacting host health through diverse pathways [60]. Furthermore, *Eysipelatoclostridium* has been implicated in developing necrotizing enterocolitis in preterm infants [61]. *Parabacteroides*, an antibiotic-resistant harmful bacterium in the gut, proliferates extensively when the host’s gut microbiota is imbalanced, leading to diseases such as appendicitis, peritoneal inflammation, and bacteremia [62]. *Escherichia-Shigella*, a type of Enterobacteriaceae, induces intestinal inflammation by penetrating epithelial cells, causing macrophage apoptosis, and releasing IL-1β [63]. In addition, Escherichia-Shigella was also dramatically overrepresented in diarrhea in calves and adult yaks [27]. The increased abundance of these bacterial genera in the diarrheic group suggests a potential association with gastrointestinal disorders, inflammation, and other related health issues.

To identify how an alteration of gut microbiota influenced diarrhea, the microbial function was predicted via a PICRUSt analysis. It is worth mentioning the activities linked to amino acid metabolism, which includes the metabolism of lysine, serine, threonine, alanine, aspartate, and glutamate, were abundant in the diarrhea group, which might be due to the active amino acid metabolism during the development of diarrhea. In addition, oxidative phosphorylation was significantly enriched in the diarrhea group, the primary pathway for generating reactive oxygen species (ROS) intermediates such as lipid peroxides, nitric oxide, and superoxide radicals [64,65]. This observation implies that a substantial amount of oxidative stress occurs in the diarrhea group. This phenomenon may partly elucidate the lower microbial diversity and the presence of intestinal inflammation in the diarrhea groups.

## 5. Conclusions

Juvenile yak diarrhea is associated with variance in the fecal microbial structure, diversity, and a reduction in the relative abundance of beneficial bacteria. The fecal microbiota in diarrheic juvenile yaks is characterized by low diversity and is dominated by Bacteroides, Parabacteroides, and Escherichia-Shigella. Dysbiosis in the fecal microbial community structure is associated with changes in the predictive metagenomic function of the bacterial communities. Therefore, some potential strategies, e.g., probiotic supplementation, dietary adjustments to support gut health, and targeted antimicrobial treatments against identified pathogens, are needed to overcome this issue in the region. 

## Figures and Tables

**Figure 1 animals-14-01181-f001:**
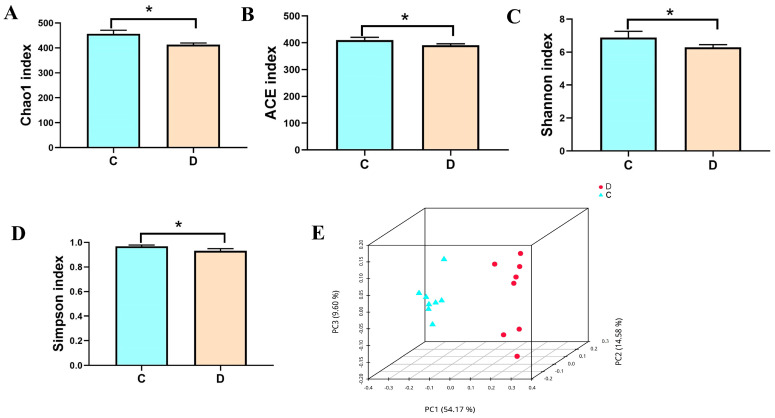
Comparative analysis of the gut microbial diversity between the C and D groups. (**A**–**D**) represent Chao, ACE, Shannon, and Simpson indexes, respectively. (**E**) Scatterplot from PCoA. C: control group, D: diarrhea group. Data are presented as the mean ± SD. * *p* < 0.05.

**Figure 2 animals-14-01181-f002:**
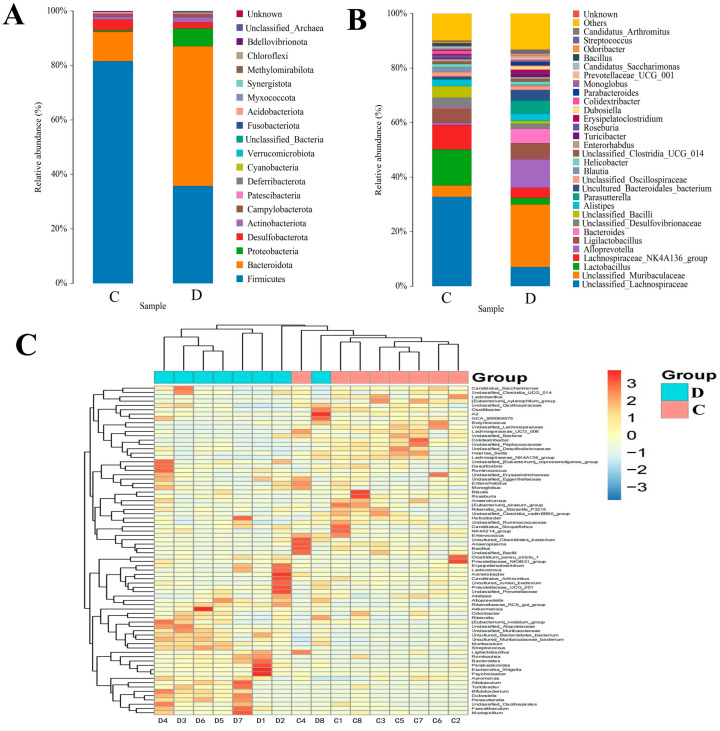
Distribution of the relative abundances of gastrointestinal microbial genera and phyla (**A**,**B**) identified in groups C and D. (**C**) Heatmap of the hierarchical microbial community clustering at the genus level in the control and diarrheal yaks. C: control group, D: diarrhea group.

**Figure 3 animals-14-01181-f003:**
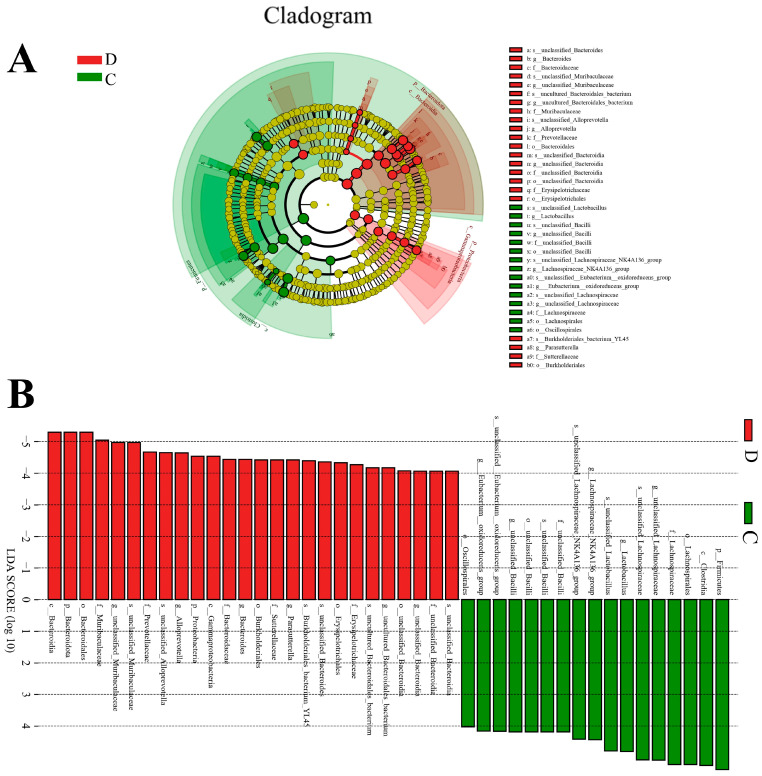
Identification of distinct biomarkers in the gut microbiota of yaks that are linked to the occurrence of diarrhea. (**A**) The cladogram shows the differential microbial phylogenetic distribution. (**B**) Variations in the microbial abundances that exist between the C and D groups. C: control group, D: diarrhea group.

**Figure 4 animals-14-01181-f004:**
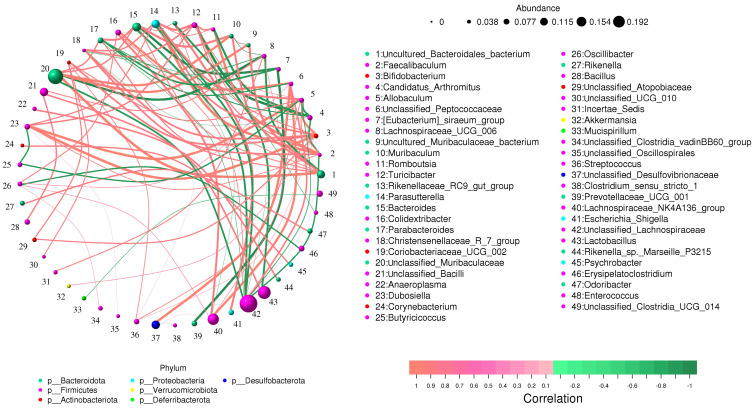
Network analysis reveals a relationship between several microbes. The orange lines represent a positive connection, whereas the green lines show a negative correlation. C: control group, D: diarrhea group.

**Figure 5 animals-14-01181-f005:**
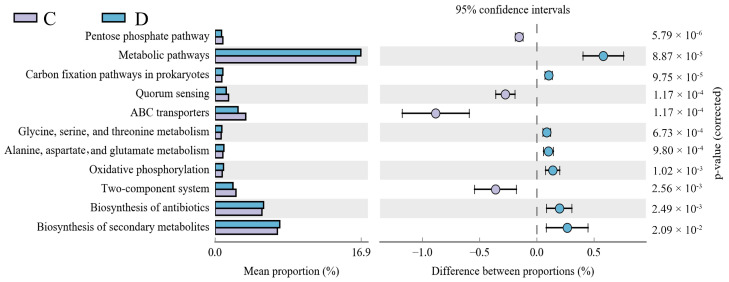
Functional prediction of the KEGG pathways using a PICRUSt analysis. Distinction-enriched pathways between the diarrhea and control groups were examined using the Kruskal-Wallis H test. C: control group, D: diarrhea group.

**Table 1 animals-14-01181-t001:** Sample sequence information, C: control group, D: diarrhea group.

Sample ID	Raw Reads	Clean Reads	Effective Reads	AvgLen(bp)	Effective (%)
C1	79,966	79,834	78,775	414	98.51
C2	79,069	78,924	78,034	413	98.69
C3	79,930	79,798	78,106	412	97.71
C4	80,082	79,932	78,858	413	98.47
C5	80,162	79,980	77086	415	96.16
C6	69,034	68,913	67,730	416	98.11
C7	57,481	57,336	56,588	415	98.45
C8	79,998	79,865	78,867	418	98.59
D1	79,966	79,834	787,75	415	98.51
D2	80,097	79,944	79,196	414	98.87
D3	79,971	79,827	78,281	415	97.89
D4	79,836	79,689	78,560	414	98.40
D5	79,979	79,825	78,529	413	98.19
D6	80,029	79,874	78,653	415	98.28
D7	64,354	64,235	63,175	416	98.17
D8	78,158	78,008	76,781	415	98.24

## Data Availability

The original sequence data were submitted to the Sequence Read Archive (SRA) (NCBI, USA) with accession no. PRJNA1089830.

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
