# Peer review of "Metagenomic Analysis Reveals A Gut Microbiota Structure and Function Alteration between Healthy and Diarrheic Juvenile Yaks"

_animals, 2024, doi:10.3390/ani14081181_

Round 1

Reviewer 1 Report

Comments and Suggestions for Authors

The goal of the study was to analyze the differences of the intestinal bacteria of Yaks of the Qinghai-Tibet Plateau region under health and disease (diarrheic) conditions.

Title: Metagenomic Analysis Reveals Gut Microbiota Structure and Function Alteration between Healthy and Diarrheic Juvenile Yaks

Abstract: The conclusion of the abstract is general and unclear.

Keywords: Consider changing the word “plateau”

Introduction: Please give more detail on the findings available from the references since there are at least 10 of them about microbiota in yaks and it is not clearly exposed which goals on the identification of the microbiata in yaks under diarrheic conditions have not been previously reached.

Materials and methods

Information on the general conditions of the place, nutrition, age, gender, breed (Bos grunniens), and genealogy (if some of the animals were related or belonged to the same family) of the sampled healthy and diarrheic animals is missing. Please mention if the causing agent of the diarrhea was diagnosed by a professional veterianarian, as well as the infection stage each animal was (incubation stage with no clinical signs or all animals presented clinical signs of diarrhea) and how many days of diarrheic evolution each animal had.

It is important to identify what kind of diarrhea was presented by the animals to have evidence that the results can be representative of one origin, because, as it was mentioned in the introduction, diarrhea can have a multicausal origin and this can intoduce a bias to the study and the interpretation of the results.

Were the samples taken before the animals received any treatment?

Results

It’s necessary to check with more detail some of the sections that could be biased as described in the comments section.

Discussion

Please check more references for this section considering that comparable information is being generated here. Moreover, more similitudes could be reported by comparing results with other works.

Please consult the following reference for more details on the microbial groups present in human under diarrheic conditions: The, H. C., & Le, S. N. H. (2022). Dynamic of the human gut microbiome under infectious diarrhea. Current Opinion in Microbiology, 66, 79-85.

For more information on the mechanism of action identified in relation to the microbiome on the gut barrier in calves and the nutritional and immune function of the gut microbiome, please consult the following reference: Du, Y., Gao, Y., Hu, M., Hou, J., Yang, L., Wang, X., ... & Xu, Q. (2023). Colonization and development of the gut microbiome in calves. Journal of Animal Science and Biotechnology, 14(1), 46.

It would also be desirable to have more comparable information from other publications in ruminants that clearly validate the microbial groups identified in the diarrheic state in ruminants such as:

Li, W., Yi, X., Wu, B., Li, X., Ye, B., Deng, Z., ... & Zhou, Z. (2023). Neonatal Calf Diarrhea Is Associated with Decreased Bacterial Diversity and Altered Gut Microbiome Profiles. Fermentation, 9(9), 827

and

Liu, J., Wang, X., Zhang, W., Kulyar, M. F. E. A., Ullah, K., Han, Z., ... & Li, K. (2022). Comparative analysis of gut microbiota in healthy and diarrheic yaks. Microbial Cell Factories, 21(1), 1-15.

Conclusions

The purpose of the generated information could be reformulated, complemented, and exposed more clearly.

References

Homogenize the formatting style since some have the journal name abbreviated, the year in bold formatting, and some have a doi.

Comments

Line 128: Include the abbreviation to identify the Control group (C) and Diarrhea group (D) in the text, tables, and figures.

Lines 141-144: Include the description for Figure 1E in the description of Figure 1.

Lines 151-152: Check and correct the units of the Simpson index, which are between 0 and 1.

Line 162: Delete the D from the parenthesis, since in the previous line Figure D is already mentioned. Delete the F from the (D, F) and change it to E because there is no Figure 1F.

Lines 202-207: Check the text and Supplement Table 1. Eighteen genera are mentioned but, according to Supplement Table 1, [Acetivibrio] _ethanolgignens_group, UCG_003 did not increase the relative abundance of bacterial genera. Besides, the taxa (UCG_005, unclassified_Anaerovoracaceae, unclassified_Clostridia, unclassified_UCG_010, Rikenella y Family_XIII_UCG_001) in which there was an increase are not mentioned. On the other hand, the following taxa are not presented in Supplement Table 1: Bacillus, Limosilactobacillus, [Eubacterium]_ventriosum_group, Leuconostoc, Fusobacterium, unclassified_Lactobacillales, unclassified_Beijerinckiaceae, Candidatus_Udaeobacter and Dialister.

Line 380: Citations [55, 56] are mentioned; however, the corresponding references are not included.

Author Response

Dear Reviewer:

Thank you for your comments on our manuscript “Metagenomic Analysis Reveals Gut Microbiota Structure and Function Alteration between Healthy and Diarrheic Juvenile Yaks” (Animals-2899871). The comments are constructive and helpful in improving our manuscript. The leading corrections and the responses to the comments are as follows (the replies are highlighted in blue).

Response to Reviewer’s Comments  

Reviewer #1: The study aimed to analyze the differences in the intestinal bacteria of Yaks of the Qinghai-Tibet Plateau region under health and disease (diarrheic) conditions.

Title: Metagenomic Analysis Reveals Gut Microbiota Structure and Function Alteration between Healthy and Diarrheic Juvenile Yaks

Abstract: The conclusion of the abstract is general and unclear.

Response: Thanks for your comments. We have made modifications to the abstract conclusion; it reads now:

“Taken together, our study showed that the harmful bacteria increased and beneficial bacteria decreased significantly in the gut microbiota of diarrhea yaks. Moreover, these results also indicated that the dysbiosis of gut microbiota may be a significant driving factor of diarrhea in yaks.”

Keywords: Consider changing the word “plateau”

Response: Thanks for your suggestion. "Plateau" has been replaced with "16sRNA".

Introduction: Please give more detail on the findings available from the references since there are at least 10 of them about microbiota in yaks and it is not clearly exposed which goals on the identification of the microbiata in yaks under diarrheic conditions have not been previously reached.

Response: Thanks for your comments. We have added relevant information on yak gut microbiota in the introduction section. It reads now:

“Thus, in contrast to poultry and other mammals, the importance of the gut microbiota in numerous physiological activities is more prominent in yaks. Zhang et al. demonstrated that the predominant bacterial phyla in the gut of healthy yaks were Firmicutes, Bacteroidetes, and Proteobacteria. A substantial presence of anaerobes from families such as Peptostreptococcaceae, Prevotellaceae, Flavobacteriaceae, Lachnospiraceae, and Succinivibrionaceae was observed across different intestinal segments [26]. Analysis of fungal microbial diversity revealed significant differences in 23 genera, including Paraconiothyrium, Monascus, Plenodomus, Phaeoacremonium, Preussia, and Mortierella, between healthy and diarrheal adult yaks from the Gannan region of Gansu Province, China [27]. Furthermore, examination of intestinal bacteria indicated a lower relative abundance of Firmicutes, Bacteroidetes, and Verrucomicrobia in diarrheal adult yaks compared to their healthy counterparts [28]. Nevertheless, the specific composition and functional alterations of intestinal bacteria in diarrheic yaks, particularly juvenile calves, remain unclear. The neonatal period in yak calves is critical for establishing the gut microbial community, as the composition formed during this period significantly influences the structure of the microbiota in later adulthood. Importantly, the gut microbiota's configuration during this stage is highly adaptable, with enduring effects on gut function, including susceptibility to diarrhea, extending into later life. Hence, this study investigates alterations in gut microbiota structure and function between healthy and diarrheic juvenile yaks, offering insights that could inform the development of preventive and therapeutic strategies for diarrhea from a gut microbial perspective.”.

Materials and methods

Information on the general conditions of the place, nutrition, age, gender, breed (Bos grunniens), and genealogy (if some of the animals were related or belonged to the same family) of the sampled healthy and diarrheic animals is missing. Please mention if the causing agent of the diarrhea was diagnosed by a professional veterinarian, as well as the infection stage each animal was (incubation stage with no clinical signs or all animals presented clinical signs of diarrhea) and how many days of diarrheic evolution each animal had.

It is important to identify what kind of diarrhea was presented by the animals to have evidence that the results can be representative of one origin because, as it was mentioned in the introduction, diarrhea could have a multicausal origin and this can introduce a bias to the study and the interpretation of the results.

Were the samples taken before the animals received any treatment?

Response: Thanks for your comments. We have modified the material method's animal and sample collection section and provided relevant information on experimental yaks in Supplementary Table 1. Now it as:

“The eight healthy and eight diarrheal yaks from the Hongyuan Yak Breeding Base in Sichuan, China (located in the eastern part of the Qinghai-Tibet Plateau at an altitude above 3500 meters) were selected, having the same immunological history. All yaks have been vaccinated against Brucella and foot-and-mouth disease but not for enterotoxigenic Escherichia coli and bovine coronavirus antibodies. All yaks provide the same amount of green grass and supplement 0.5 kg of purified diet daily (corn meal, bran, soybean meal, NaCHO3, etc., in proportion to calves' nutritional requirements). Before collecting samples, professional veterinarians evaluate and analyze the health status of yaks and determine that pathogenic infections did not cause diarrhea in yaks and have not been treated. When the diarrheal group presented clinical signs of diarrhea. The rectum was swabbed with sterile swabs in a rotating way. The obtained samples were immediately placed in sterile plastic containers, transported to the lab, and stored at -80°C for future investigation.”

Results

It’s necessary to check with more detail some of the sections that could be biased as described in the comments section.

Response: Thank you for your comment. We have made modifications to the ambiguous results section to present it more clearly.

Discussion

Please check more references for this section considering that comparable information is being generated here. Moreover, more similitudes could be reported by comparing results with other works.

Please consult the following reference for more details on the microbial groups present in human under diarrheic conditions: The, H. C., & Le, S. N. H. (2022). Dynamic of the human gut microbiome under infectious diarrhea. Current Opinion in Microbiology, 66, 79-85.

For more information on the mechanism of action identified in relation to the microbiome on the gut barrier in calves and the nutritional and immune function of the gut microbiome, please consult the following reference: Du, Y., Gao, Y., Hu, M., Hou, J., Yang, L., Wang, X., ... & Xu, Q. (2023). Colonization and development of the gut microbiome in calves. Journal of Animal Science and Biotechnology, 14(1), 46.

It would also be desirable to have more comparable information from other publications in ruminants that clearly validate the microbial groups identified in the diarrheic state in ruminants such as:

Li, W., Yi, X., Wu, B., Li, X., Ye, B., Deng, Z., ... & Zhou, Z. (2023). Neonatal Calf Diarrhea Is Associated with Decreased Bacterial Diversity and Altered Gut Microbiome Profiles. Fermentation, 9(9), 827 and Liu, J., Wang, X., Zhang, W., Kulyar, M. F. E. A., Ullah, K., Han, Z., ... & Li, K. (2022). Comparative analysis of gut microbiota in healthy and diarrheic yaks. Microbial Cell Factories, 21(1), 1-15.

Response: Thank you for your suggestion. We have added relevant information on changes in gut microbiota of diarrhea animals in the discussion.

“Compared to the healthy group, we observed a significant reduction in the variety and richness of gut microbiota in the diarrheal yaks. This indicates the presence of gut bacterial dysbiosis. Consistent with our investigations, Li et al. profiled the fecal microbial community of 21 calves with varying health conditions using the 16S rRNA gene and also found that the diversity and evenness index of diarrhea calves were significantly reduced [31]. The detection of gut microbiota structure of adult yaks with diarrhea also presents a distinct dec [28] line in alpha diversity, accompanied by significant shifts in taxonomic compositions. In addition, Wang et al. found that diarrheic Boer goats had much lower gut bacterial diversity than healthy herds [15]. Furthermore, girafes' gut bacterial diversity decreased during diarrhea [22]. Juvenile animals' gut microbiome community structures are volatile and quickly change by species, genotype, environment, and nutrition during development, eventually stabilizing at maturity [32,33]. Research has shown that maintaining a proper gut bacteria composition and diversity is necessary for performing complicated physiological processes. Conversely, intestinal microbiome disturbance is identified as a central or driving factor in various diseases, including enteritis and diarrhea [33]. Dysbiosis of the intestinal microbiota has been connected to diarrheal yaks, marked by increased mortality and weight loss.

Previous research suggests that gut microbial dysbiosis can affect human immunity and intestinal permeability, increasing vulnerability to pathogens. Additionally, specific pathogens may exhibit heightened pathogenicity in the context of intestinal flora disturbance. Consequently, we speculate that the increased mortality and weight loss in diarrheic yaks may be related to gut microbial dysbiosis. This study found that phyla Firmicutes and Bacteroidetes were the most common bacteria in the intestines of a healthy yak calf. In line with our results, the prevalence of these two phyla was noted in the intestinal tracts of sheep, cattle, and steers, underscoring the ecological and functional significance of these dominant phyla in the gastrointestinal tract of ruminants [34–36]. Firmicutes encompass many gram-positive bacteria, including beneficial species such as Lactobacillus, Lactococcus, and Listeria. These bacteria are crucial in maintaining a balanced gut microbiota and preventing pathogenic invasions [37,38]. In ruminants, Firmicutes is mainly involved in the digestion of fiber and cellulose, contributing to overall digestive processes [39].”

Conclusions

The purpose of the generated information could be reformulated, complemented, and exposed more clearly.

Response: Thank you for your suggestion. We have made modifications to the conclusion, now it as:

“In summary, Juvenile yak diarrhea is associated with variance in fecal microbial structure and diversity and a reduction in the relative abundance of beneficial bacteria. The fecal microbiota with diarrheic Juvenile yaks were characterized by low diversity and were dominated by Bacteroides, Parabacteroides, and Escherichia-Shigella. Dysbiosis in fecal microbial community structure was associated with calf diarrhea and changes in the predictive metagenomic function of the bacterial communities.”

References

Homogenize the formatting style since some have the journal name abbreviated, the year in bold formatting, and some have a doi.

Response: Thank you for your suggestion. We have standardized the format of the references.

Comments

Line 128: Include the abbreviation to identify the Control group (C) and Diarrhea group (D) in the text, tables, and figures.

Response: Thanks for your suggestion. We have made changes to the manuscript.

Lines 141-144: Include the description for Figure 1E in the description of Figure 1.

Response: Respected reviewers, we added the description of Figure 1E to Figure 1.

Lines 151-152: Check and correct the units of the Simpson index, which are between 0 and 1.

Response: Thanks for your comments. The Simpson index in our manuscript is between 0 and 1, which is correct. To address your concerns, we have provided some publications for your kind consideration.

  • Lei Wang, Fazul Nabi, (2024). Low-dose thiram exposure elicits dysregulation of the gut microbial ecology in broiler chickens. Ecotoxicology and Environmental Safety, 270: 115879. https://doi.org/10.1016/j.ecoenv.2023.115879
  • Liu, Z., Li, A., Wang, Y , (2020). Comparative analysis of microbial community structure between healthy and Aeromonas veronii-infected Yangtze finless porpoise. Microbial Cell Factories,19:123. https://doi.org/10.1186/s12934-020-01383-4

Line 162: Delete the D from the parenthesis since Figure D is already mentioned in the previous line. Delete the F from the (D, F) and change it to E because there is no Figure 1F.

Response: We apologize for our misdescription; we have modified this error.

Lines 202-207: Check the text and Supplement Table 1. Eighteen genera are mentioned but according to Supplement Table 1, [Acetivibrio] _ethanolgignens_group, UCG_003 did not increase the relative abundance of bacterial genera. Besides, the taxa (UCG_005, unclassified_Anaerovoracaceae, unclassified_Clostridia, unclassified_UCG_010, Rikenella y Family_XIII_UCG_001) in which there was an increase are not mentioned. On the other hand, the following taxa are not presented in Supplement Table 1: Bacillus, Limosilactobacillus, [Eubacterium]_ventriosum_group, Leuconostoc, Fusobacterium, unclassified_Lactobacillales, unclassified_Beijerinckiaceae, Candidatus_Udaeobacter and Dialister.

Response: We apologize for our mistake; we have rechecked and added the missing bacterial genera in supplement Table 2.

Line 380: Citations [55, 56] are mentioned; however, the corresponding references are not included.

Response: We are sorry for our carelessness. We have added relevant references in the manuscript.

Reviewer 2 Report

Comments and Suggestions for Authors

Intestinal bacteria comprise along with  viruses and mycotic organisms the intestinal microbiome.  The bacteria are best studied, but not in all species, this study from an agricultural study group focuses exclusively on the novel yak.  Therefore it has importance and their standard microbiome should be compared a bit more rigorously with other herd agricultural species and speculation on why yak may have some unique features.  The investigations are whether dysbiosis or changes in the bacteria may either cause or change as a result of diarrhea, a common lethality factor for yak.  It should be described in a bit more detail what the nature of this diarrhea is. I suspect it may be inflammatory and bacterial pathogen in nature, but it could be genetic suspceptibility.  A  significant strength is the metabolomics analysis with diarrhea which broadens the impact of the data. Importantly for this PICRUST analysis as well as for the bacterial changes with diarrhea the authors have been conservative in their interpretation.  To prove that one or a group of bacteria are responsible and fulfill Koch’s postulate in a whole animal diarrhea model without germ free subjects may be impossible.  The investigations deal with the real world conditions and the changes are discussed intelligently with background provided why certain bacteria might be responsible. Therefore this manuscript has a straightforward hypothesis needs some text addition but no experimentation. 

Comments on the Quality of English Language

very good, only a few instances of unusual syntax

Author Response

Dear Reviewer:

Thank you for your comments on our manuscript “Metagenomic Analysis Reveals Gut Microbiota Structure and Function Alteration between Healthy and Diarrheic Juvenile Yaks” (Animals-2899871). The comments are constructive and helpful in improving our manuscript. The leading corrections and the responses to the comments are as follows (the replies are highlighted in blue).

Reviewer #2: Intestinal bacteria comprise along with viruses and mycotic organisms the intestinal microbiome.  The bacteria are best studied, but not in all species, this study from an agricultural study group focuses exclusively on the novel yak.  Therefore, it has importance and their standard microbiome should be compared a bit more rigorously with other herd agricultural species and speculation on why yak may have some unique features.  The investigations are whether dysbiosis or changes in the bacteria may either cause or change as a result of diarrhea, a common lethality factor for yak.  It should be described in a bit more detail what the nature of this diarrhea is. I suspect it may be inflammatory and bacterial pathogen in nature, but it could be genetic suspceptibility.  A significant strength is the metabolomics analysis with diarrhea which broadens the impact of the data. Importantly for this PICRUST analysis as well as for the bacterial changes with diarrhea the authors have been conservative in their interpretation.  To prove that one or a group of bacteria are responsible and fulfill Koch’s postulate in a whole animal diarrhea model without germ free subjects may be impossible.  The investigations deal with the real world conditions and the changes are discussed intelligently with background provided why certain bacteria might be responsible. Therefore, this manuscript has a straightforward hypothesis needs some text addition but no experimentation. 

Response: Thank you for your thoughtful review of our manuscript. We appreciate your suggestion to compare the yak microbiome with other agricultural species and provide more details on the nature of yak diarrhea. We have modified this in our Front and Discussion and compared and analyzed the similarity of gut microbiota changes in yak diarrhea with diarrhea in other cattle species. In the next step, we will carry out a metabolomics study on the intestinal contents of diarrheal yak based on your suggestion and analyze the intestinal metabolites with other cattle species. In addition, the use of PICRUST to analyze the functional changes of intestinal flora provides more information between diarrhea and intestinal flora, and can reveal the functional changes of bacterial flora in diarrhea. It has been widely used to analyze the functional changes of intestinal flora in other species. So, we referred to this approach. To address your concerns, we have provided some references for your kind consideration.

  • Li W, Yi X, Wu B, et al. Neonatal Calf Diarrhea Is Associated with Decreased Bacterial Diversity and Altered Gut Microbiome Profiles. Fermentation. 2023; 9(9):827. https://doi.org/10.3390/fermentation9090827
  • Kim, E.-T, Lee, S.-J, et al. Dynamic Changes in Fecal Microbial Communities of Neonatal Dairy Calves by Aging and Diarrhea. Animals,2021;11:1113. https://doi.org/10.3390/ani11041113
  • Ruvalcaba-Gómez, J.M. Villaseñor-González, F, Espinosa-Martínez, M.A. et al. Growth Performance and Fecal Microbiota of Dairy Calves Supplemented with Autochthonous Lactic Acid Bacteria as Probiotics in Mexican Western Family Dairy Farming. Animals. 2023; 13: 2841. https://doi.org/10.3390/ani13182841

Reviewer 3 Report

Comments and Suggestions for Authors

Suggested revisions:

Abstract

No comments

Introduction

Line 36: typographical error at “oreover,”

Line 46, 48 and 49: change flora to microbiome, this is outdated terminology

Line 49: the microbiome changes for the conditions described in this sentence are narrowed or reduced diversity. More description on how the microbiome changes is needed here.

Line 54: “microbiotas” should not be pluralized, this should be “microbiota”

Line 54: grammar check, should be “particularly affecting vulnerable gut microbiota in lambs and piglets [13–15].”

Line 61: change gut flora to intestinal microbiome

Line 69-71: “However, how yaks and diarrhea gut microbiota could be related is still unclear. This research examines the differences and compositions of gut bacteria in healthy and diarrheal yaks.” These two statements are unclear and need to be rewritten. Firstly, “how yaks and diarhhea gut microbiota could be related…” is non-sensical. You want to determine if dysbiosis in the gut microbiome causes diarrhea in yaks. Second, “This research examines the differences and compositions of gut bacteria” is not effectively stating what was done in this research. The research attempted to characterize the differences between gut community compositions of health and diarrheal yaks.

Materials and Methods

Line 74: It is not clear in the methods or elsewhere in the manuscript if there were 8 yaks used in total (4 healthy, 4 with diarrhea) or 8 yaks per group. This needs to be clarified.

Results

Line 126: 16 fecal samples analyzed, 8 yaks mentioned in methods. This is leading author to assume that there were 8 yaks per group, or samples were run in duplicate. Please clarify.

Line 139: Table 1 title needs to be moved so it is not split over 2 pages

Line 147: change “gut flora” to gut microbiome community

Discussion

Line 278:  the double reference is not needed at the end of this sentence “Li et al. [18].”

Line 279: change gut flora to “gut microbiome community structure”

Line 283: change intestinal flora to intestinal microbiome

Line 307: “harmful production of LPS”, all gram negative bacteria have LPS in the outer membrane, they are producing more with expansion of the population. It would be more accurate to state “Moreover, an expansion in the Proteobacteria population leads to increased production of lipopolysaccharides (LPS), which can lead to intestinal inflammation in the host [35].

Line 343-344: “Reducing beneficial bacteria in diarrheic yaks is crucial for preserving host health and intestinal homeostasis.” This statement is non-sensical. It should be increasing and not reducing. Rephrase the ending of this paragraph to read more clearly. Additionally, the terms beneficial/harmful bacteria does not apply to the intestinal microbiome structure, it is known that each species has a niche role within the community, and we are learning how the balance is important in maintaining health. What could be harmful in one sense can be beneficial in another. Diarrhea may not be due to the invasion of a new pathogenic bacterial species, it could be due to an imbalance of the community structure.

Line 387: remove “beneficial bacteria” from this summary. Convey the message that shifts which narrowed/decrease populations or expand bacteria originally less prevalent within the microbiome community structure disrupts the balance of its functional roles, leading to diarrhea and health impacts over time.

Supplemental Data

No comment

Comments on the Quality of English Language

Some grammatical errors have been noted in feedback above.

Author Response

Dear Reviewer:

Thank you for your comments on our manuscript “Metagenomic Analysis Reveals Gut Microbiota Structure and Function Alteration between Healthy and Diarrheic Juvenile Yaks” (Animals-2899871). The comments are constructive and helpful in improving our manuscript. The leading corrections and the responses to the comments are as follows (the replies are highlighted in blue).

Reviewer #3

Introduction

Line 36: typographical error at “oreover,”

Line 46, 48 and 49: change flora to microbiome, this is outdated terminology

Line 49: the microbiome changes for the conditions described in this sentence are narrowed or reduced diversity. More description on how the microbiome changes is needed here.

Line 54: “microbiotas” should not be pluralized, this should be “microbiota”

Line 54: grammar check, should be “particularly affecting vulnerable gut microbiota in lambs and piglets [13–15].”

Line 61: change gut flora to intestinal microbiome

Line 69-71: “However, how yaks and diarrhea gut microbiota could be related is still unclear. This research examines the differences and compositions of gut bacteria in healthy and diarrheal yaks.” These two statements are unclear and need to be rewritten. Firstly, “how yaks and diarhhea gut microbiota could be related…” is non-sensical. You want to determine if dysbiosis in the gut microbiome causes diarrhea in yaks. Second, “This research examines the differences and compositions of gut bacteria” is not effectively stating what was done in this research. The research attempted to characterize the differences between gut community compositions of health and diarrheal yaks.

Response: Thanks for the reviewer's constructive suggestions. We have made corresponding modifications based on your suggestions. We tried our best to reorganize and rewrite introduction section of the manuscript, and we hope the revised manuscript will cover your suggested points.

Materials and Methods

Line 74: It is not clear in the methods or elsewhere in the manuscript if there were 8 yaks used in total (4 healthy, 4 with diarrhea) or 8 yaks per group. This needs to be clarified.

Results

Line 126: 16 fecal samples analyzed, 8 yaks mentioned in methods. This is leading author to assume that there were 8 yaks per group, or samples were run in duplicate. Please clarify.

Line 139: Table 1 title needs to be moved so it is not split over 2 pages

Line 147: change “gut flora” to gut microbiome community

Response: Dear reviewer, we have revised the manuscript and provided relevant weight, gender, age, and grouping information of yaks in Supplementary Table 1. We hope that the revised manuscript will cover your suggested points.

Discussion

Line 278:  the double reference is not needed at the end of this sentence “Li et al. [18].”

Line 279: change gut flora to “gut microbiome community structure”

Line 283: change intestinal flora to intestinal microbiome

Line 307: “harmful production of LPS”, all gram negative bacteria have LPS in the outer membrane, they are producing more with expansion of the population. It would be more accurate to state “Moreover, an expansion in the Proteobacteria population leads to increased production of lipopolysaccharides (LPS), which can lead to intestinal inflammation in the host [35].

Line 343-344: “Reducing beneficial bacteria in diarrheic yaks is crucial for preserving host health and intestinal homeostasis.” This statement is non-sensical. It should be increasing and not reducing. Rephrase the ending of this paragraph to read more clearly. Additionally, the terms beneficial/harmful bacteria does not apply to the intestinal microbiome structure, it is known that each species has a niche role within the community, and we are learning how the balance is important in maintaining health. What could be harmful in one sense can be beneficial in another. Diarrhea may not be due to the invasion of a new pathogenic bacterial species, it could be due to an imbalance of the community structure.

Line 387: remove “beneficial bacteria” from this summary. Convey the message that shifts which narrowed/decrease populations or expand bacteria originally less prevalent within the microbiome community structure disrupts the balance of its functional roles, leading to diarrhea and health impacts over time.

Response: Respected reviewer. We have made corresponding modifications based on your suggestions (lines 364-365, 409-415, 450-456), and we hope the revised manuscript will cover your suggested points.

Reviewer 4 Report

Comments and Suggestions for Authors

Comments on the Quality of English Language

Moderate editing of English language required

Author Response

Dear Reviewer:

Thank you for your comments on our manuscript “Metagenomic Analysis Reveals Gut Microbiota Structure and Function Alteration between Healthy and Diarrheic Juvenile Yaks” (Animals-2899871). The comments are constructive and helpful in improving our manuscript. The leading corrections and the responses to the comments are as follows (the replies are highlighted in blue).

Reviewer #4: Investigating changes in the animal gut microbiota with different infection caused illness? For example diarrhea is presented significant interests. The authors of “Metagenomic Analysis Reveals Gut Microbiota Structure and Function Alteration between Healthy and Diarrheic Juvenile Yaks” article demonstrated a considerable drop in the alpha diversity of the gut microbiota in diarrheic yaks, accompanied by major alterations in taxonomic compositions. According to authors, at the phylum level, the diarrhea yaks exhibited a higher abundance of Bacteroidota and Proteobacteria than healthy groups, while the level of Firmicutes decreased. Also was found that PICRust analysis verified the elevation of gut-microbial metabolic pathways in diarrhea groups, including glycine, serine, and threonine metabolism, alanine, aspartate, oxidative phosphorylation, glutamate metabolism, antibiotic biosynthesis, and secondary metabolite biosynthesis. However, the issue of influence of key some pathogenic and opportunistic bacteria in the animal gut microbiota is not new and it has been widely studied, but it is not mentioned by the authors themselves. Despite this, the results are quite interesting but still the flaws are needed to be addressed. Therefore, the paper needs major revision - specific comments:

Response: We thank the reviewer for the constructive feedback. We have tried our best to reorganize and rewrite our manuscript and hope it is more concise and understandable.

“Diarrhea is a primary cause of decreased production performance and mortality in livestock, substantially influencing the livestock industry’s development worldwide [12]. Diarrhea is mainly attributed to infectious and non-infectious factors, but regardless of which factor causes diarrhea, it can disrupt the host's gut microbiota. Extensive research has highlighted the prevalence of diarrhea across various mammalian species, particularly affecting vulnerable gut microbiota in lambs and piglets [13–15]. Imbalance in gut microbiota has been linked to numerous gastrointestinal disorders, including diarrhea and irritable bowel syndrome.  Many studies have underscored gut microbiota's pivotal role in preventing, controlling, and diagnosing diarrhea [16,17]. Gut microbiota plays a protective role in pathogen defense, as dysbiosis of the intestinal microbiota can affect various microbial functions related to nucleotide transport and metabolism, defense, translation, and transcription [18]. Furthermore, reduced gastrointestinal barrier function and disruptions in the gut microbiota are associated with diarrhea and alterations in the gut microbiota [19]. Moreover, diarrhea and gut microbiota disruptions have a bidirectional relationship [20]. Specifically, diarrhea disrupts the intestinal microbiome balance, while an imbalance in the intestinal microbiota due to high-value-added exogenous pathogens can also lead to diarrhea. Diarrheic calves exhibited reduced gut microbiota diversity and significant changes in fecal microbial composition compared to healthy calves [21]. Wang et al. revealed notable changes in the intestinal microbiota of diarrheal goats, together with higher death rates [15]. Similarly, Li et al. discovered significant alterations in giraffes’ intestinal microbiome after episodes of diarrhea [22]. Furthermore, diarrhea is characterized by a decrease in beneficial bacteria that produce short-chain fatty acids (SCFAs), which play a role in reducing the risk of diarrhea.”

  1. Line 36. Oreover > Maybe Moreover?

Response: Thanks for your suggestion. We changed “Oreover " to "Moreover "(line 36)

  1. Line 76. Having the same immunological history. Which immunological history exactly was mentioned? Can these data be used in this research?

Response: Thank you for your comment. We have provided relevant immunological information in the manuscript methodology.

  1. The Journal is dedicated to Animal sciences. Hence I suggest to authors add some physical (weight, etc) properties of selected animals and try to link these properties obtained results of gut microbiota.

Response: Thank you for your suggestion. The aim of this study is to reveal the changes in gut microbiota and function during calf yak diarrhea. Our next step is to conduct metabolomics research based on your suggestion, with a focus on investigating the correlation between gut microbiota and its metabolites on animal weight and immune indices.

  1. The link to used primers of 16S rRNA is missing.

Response: Dear reviewer, currently the 16sRNA sequencing region of the gut bacteria microbiota is V3-V4, so we have chosen the universal primer 338F, 5 '- ACTCCTACGGGGGGCAGCA-3' for the V3-V4 region; And 806R, 5 '- GACTACHVGGTWTTCTAAT-3') We have provided relevant information in the manuscript. The following references are provided for your kind consideration.

  • Wang Y, An M, Zhang Z, Zhang W, Kulyar MF, Iqbal M,He Y, Li F, An T, Li H, Luo X, Yang S, Li J, 2022. Effects of Milk Replacer-Based Lactobacillus on Growth and Gut Development of Yaks’ Calves: a Gut Microbiome and Metabolic Study. Microbiol Spectr 10:e01155-22.https://doi.org/10.1128/spectrum.01155-22

  1. Also were not cited all used software in section of Bioinformatics and Statistical Analysis.

Response: Thanks for your suggestion. I have provided the relevant software used in Bioinformatics and Statistical Analysis.

5.1. Were paired-end reads were merged? Which software was used for this purposes?

Response: Respected Reviewer, paired-end sequences were merged using PEAR v0.9.1016 with default parameters. We have provided detailed information in the manuscript.

  1. After clustering OTUs, they are should be taxonomically assigned. Which database was used for taxonomically assignment? Do cite the used database.

Response: Respected reviewer, representative sequence of each OTU was taxonomically classified based on the Ribosomal Database Project (RDP) database. The phylogenetic relationships of different OTUs, and multiple sequence alignments were conducted using the MUSCLE software. We have provided detailed information in the manuscript.

  1. Same with PICRUSt software.

Response: We have provided detailed information in the manuscript. Now it can read as:

“KEGG Ortholog function of bacteria micro population in yaks was predicted using PICRUSt annotating databases of KEGG and COG. Alterations in intestinal bacteria function were analyzed through ANOVA and Dunn tests.”

  1. Lines 127-129. Are the total counts of obtained sequences correct? 122,8112? Maybe 1,228,112?

Response: We have modified this error (lines 165-168)

  1. The figures which show analysis of sequencing depth and distribution of OTUs are technical and there is nothing to do with them. Or authors were planning to discuss these results? I suggest exclude these figures or move to supplementary material.
  2. Also is suggested to attach the OTUs table (with taxonomy) in the supplementary materials.

Response: As suggested, we have move Figure 1 to supplementary material.

  1. Fig 2. C and D groups are …? Readers want to see where healthy, and where ill yaks.

Response: Dear reviewer, C group is Control group,D group is Diarrheic group. We have labelled in the manuscript and figure legends accordingly. We appreciate your opinion.

  1. Lines 173-175. The names of phyla should be capitalized.
  2. Fig.3. Improve the quality of figures (use the figure redactors). Difficult to see or read something there.
  3. Same with figure 4 and 5. Nothing is seen.

Response: Respected reviewer, we have replaced the figures with full DPI figures. Also, we have uploaded figures separately for better clarity.

  1. Figure 5. Authors did network analyses. But in the method section, this analyse is not mentioned and cited. Please, correct.

Response: Thank you for your suggestion. We have added the method of microbial network correlation analysis in the Bioinformatics and Statistical Analysis of materials and methods. Now it read as:

“Based on species abundance, each genus's correlation coefficient (Spearman Correlation coefficient) was calculated, and the correlation coefficient matrix was obtained. Cytoscape V. 3.6.1 was used to visualize networks with significant correlations between genera.”

  1. Line 146. I am a little bit skeptical about the results from PICRUSt metagenomic functional prediction. Usually, to say something about metabolism of the microbiota, there is carried out shotgun metagenome, not just 16S rRNA amplicon sequencing. Please, justify why you used PICRUSt? The results of PICRUSt are comparable with the results of shotgun metagenome sequencing? I am not sure that they are comparable.

Response: Dear Reviewer, PICRUSt2 can predict 16S rRNA gene sequences across multiple functional databases, including MetaCyc (https://metacyc.org/), KEGG (https://www.kegg.jp/), COG (https://www.ncbi.nlm.nih.gov/COG/), Pfam (http://pfam.xfam.org/), and TIGRFAM (http://tigrfams.jcvi.org/cgi-bin/index.cgi), etc. We utilized the most used KEGG and COG databases for annotation results. PICRUSt has also been widely employed for macro genomic functional predictions in other studies. Here are some references for your consideration.

  • The gut microbiome predicts response to UDCA/CDCA treatment in gallstone patients: comparison of responders and non-responders. DOI: 10.1038/s41598-024-53173-2
  • Refractory Helicobacter pylori infection and the gastric microbiota. DOI:10.3389/fcimb.2022.976710
  1. Regarding data availability. As a reviewer, I wanted to revisit your data using some methods to verify the accuracy of your results. But the sequence data has not been deposited in open public databases such as NCBI. Sequences are typically deposited in the SRA (NCBI). I suggest you to deposit your data and mention it.
    Response: Dear Reviewer, the original sequence data was submitted to the Sequence Read Archive (SRA) (NCBI, USA) with the accession no. PRJNA1089830, and mentioned in the revised manuscript (line 471-472).

  1. Conclusions have to be rewritten. This section should not just be a summary; it should contain a take-home message.

Response: Respected reviewer, we have rewritten the conclusion to express clearer information.

Hence, to recommend the publication of the article, all the above-mentioned comments should be addressed carefully, only then the manuscript be considered for acceptance. Also, the style of writing shall be improved in some places.

Response: Thanks for your comments. We tried our best to recheck the manuscript and made corresponding revisions. We hope our revision addresses all of your concerns.

Round 2

Reviewer 1 Report

Comments and Suggestions for Authors

The goal of the study was to analyze the differences of the intestinal bacteria of Yaks of the Qinghai-Tibet Plateau region under health and disease (diarrheic) conditions.

Title: Metagenomic Analysis Reveals Gut Microbiota Structure and Function Alteration between Healthy and Diarrheic Juvenile Yaks

The document was improved according to the previous revision, so only some minor comments remain.

Which preventive and therapeutic strategies for diarrhea from an intestinal microbial perspective are mentioned?

  if professional veterinarians determined that the infections

   pathogens did not cause diarrhea in the yaks so what was the cause?

  What does the S in the parentheses of Figure S1A-C mean?

Reference format review and homogenize

Author Response

Q1: Thank you for your responses to my comments. However, it is a little unclear to me why the literature you cited is only in the list of used references, but is not in the text of the article itself? Why authours did not insert links in the text, in the methods and materials of the study (e.g.  The V3-V4 region of the 116 bacterial 16S rRNA gene was amplified using universal primers (338F, 5’-117 ACTCCTACGGGAGGCAGCA-3’; and 806R, 5’-GGACTACHVGGGTWTCTAAT-3’) [29] or Cytoscape V. 3.6.1 [32])?

Response: Respected reviewer, we have revised the manuscript to address your concern. References have been added where necessary to provide direct citations for specific methods and materials mentioned in the text. Thank you for bringing this to our attention, and we appreciate your feedback.

Q2: As a reviewer, I evaluate only the scientific component of your article, however, it seemed to me a little unprofessional that the authors added two more authors from Saudi Arabia during the review, and accordingly indicated the source of funding they received. At the same time, in the Author Contributions part, the contributions of these lately added authors in the study of the microbiota of the gastrointestinal tract of yaks from the Qinghai-Tibet Plateau are not given. Please arrange this change in authorship with the editor.

Response: Respected reviewer, we appreciate your evaluation of our manuscript and your concerns regarding the addition of authors during the review process. The inclusion of Saudi Arabian collaborators was due to a longstanding lab-based collaboration, significantly enhancing our study. Their dynamic involvement, particularly in resource provision, validation, and funding acquisition, has strengthened our research. We have rectified the oversight in the Author Contributions section and provided a detailed justification to the editor with the consent of all authors. We assure you of our commitment to professionalism and transparency in our work.

Reviewer 4 Report

Comments and Suggestions for Authors

Dear authors,

Thank you for your responses to my comments. However, it is a little unclear to me why the literature you cited is only in the list of used references, but is not in the text of the article itself? Why authours did not insert links in the text, in the methods and materials of the study (e.g.  The V3-V4 region of the 116 bacterial 16S rRNA gene was amplified using universal primers (338F, 5’-117 ACTCCTACGGGAGGCAGCA-3’; and 806R, 5’-GGACTACHVGGGTWTCTAAT-3’) [29] or Cytoscape V. 3.6.1 [32])?

As a reviewer, I evaluate only the scientific component of your article, however, it seemed to me a little unprofessional that the authors added two more authors from Saudi Arabia during the review, and accordingly indicated the source of funding they received. At the same time, in the Author Contributions part, the contributions of these lately added authors in the study of the microbiota of the gastrointestinal tract of yaks from the Qinghai-Tibet Plateau are not given. Please arrnage this change in authorship with the editor. 

Author Response

The goal of the study was to analyze the differences of the intestinal bacteria of Yaks of the Qinghai-Tibet Plateau region under health and disease (diarrheic) conditions. Title: Metagenomic Analysis Reveals Gut Microbiota Structure and Function Alteration between Healthy and Diarrheic Juvenile Yaks. The document was improved according to the previous revision, so only some minor comments remain.

Q1: Which preventive and therapeutic strategies for diarrhea from an intestinal microbial perspective are mentioned?

Response: Respected reviewer, our study focus is not on preventive and therapeutic strategies for diarrhea from an intestinal microbial perspective. However, potential strategies may include probiotic supplementation, dietary adjustments to support gut health, and targeted antimicrobial treatments against identified pathogens. We have included these strategies in our manuscript to overcome such issue in the region.

Q2: If professional veterinarians determined that the infections. Pathogens did not cause diarrhea in the yaks so what was the cause?

Response: Respected reviewer, the professional veterinarians confirmed that infection is not a direct cause of diarrhea in yaks. Our results suggest that dysbiosis of the gut microbiota and weaning stress, rather than pathogenic infections, may be the significant drivers of diarrhea. Therefore, further investigation into non-infectious factors contributing to the condition is warranted.

Q4: What does the S in the parentheses of Figure S1A-C mean?

Response: Respected reviewer, the “S” in the parentheses of Figure S1A-C denotes that these figures are Supplementary Figures, providing additional supporting information beyond the main content of the manuscript.

Q5: Reference format review and homogenize.

Response: Thank you for your feedback regarding the reference format. We have endeavored to address this concern by utilizing the reference style suggested by Mendeley software. However, if any inconsistencies persist, we trust that the journal's production team will ensure homogenization of the reference format during the production process. We appreciate your attention to this matter and remain committed to meeting the journal's standards.